

# Influence of sediment characteristics on the composition of soft-sediment intertidal communities in the northern Gulf of Mexico

Kyle E. Coblentz[1], Jessica R. Henkel[2], Bryan J. Sigel[3] and Caz M. Taylor[2]

[1] Department of Integrative Biology, Oregon State University, Corvallis, OR, USA
[2] Department of Ecology and Evolutionary Biology, Tulane University, New Orleans, LA, USA
[3] Department of Physical & Life Sciences, Nevada State College, Henderson, NV, USA

## ABSTRACT

Benthic infaunal communities are important components of coastal ecosystems. Understanding the relationships between the structure of these communities and characteristics of the habitat in which they live is becoming progressively more important as coastal systems face increasing stress from anthropogenic impacts and changes in climate. To examine how sediment characteristics and infaunal community composition were related along the northern Gulf of Mexico coast, we sampled intertidal infaunal communities at seven sites covering common habitat types at a regional scale. Across 69 samples, the communities clustered into four distinct groups on the basis of faunal composition. Nearly 70% of the variation in the composition of the communities was explained by salinity, median grain size, and total organic content. Our results suggest that at a regional level coarse habitat characteristics are able to explain a large amount of the variation among sites in infaunal community structure. By examining the relationships between infaunal communities and their sedimentary habitats, we take a necessary first step that will allow the exploration of how changes in habitat and community composition influence higher trophic levels and ecosystem scale processes.

## INTRODUCTION

Coastal habitats are among the most productive and threatened on Earth and the benthic infauna, the organisms inhabiting the matrix of sediment of these habitats, are important constituents of the ecosystem. Benthic infauna are responsible for a significant amount of sediment bioturbation, secondary productivity, and nutrient cycling (*Graf & Rosenberg, 1997*). Therefore, cultivating an understanding of the links between benthic infauna and their habitats is an important step in working towards understanding how alteration and degradation of these habitats may influence the functioning of coastal ecosystems.

The soft-sediment intertidal habitats of the northern Gulf of Mexico coast provide a system for understanding community/habitat relationships and their implications

Corresponding author
Kyle E. Coblentz,
kyle.coblentz@science.oregonstate.edu

for larger scale processes. The intertidal infaunal communities along the northern Gulf of Mexico coast are relatively understudied (but see *Shelton & Robertson, 1981*; *Rakocinski et al., 1991*; *Dubois et al., 2009*) despite being an important resource for many species, including threatened migratory shorebirds, a variety of fishes and economically important invertebrates (*Gloeckner & Luczkovich, 2008*; *Henkel, Sigel & Taylor, 2012*; *Hsueh, McClintock & Hopkins, 1992*; *McTigue & Zimmerman, 1998*; *Quammen, 1984*). The habitats of these communities are experiencing unprecedented geological and anthropogenic changes. Modifications to the flow of the Mississippi River have led to drastic changes in the coastal geology of the Mississippi deltaic region and surrounding areas (*Day Jr et al., 2007*), and has also resulted in the highest rates of relative sea-level rise globally (*Penland & Ramsey, 1990*; *Zervas, 2009*). Moreover, in 2010 the region was affected by the Deepwater Horizon oil spill, the largest oil spill in United States history (*National Commission on the BP Deepwater Horizon Oil Spill and Offshore Drilling, 2011*). But understanding how infaunal communities fit into this framework of change and the potential for larger scale consequences at the ecosystem level requires a baseline understanding of the relationships of communities to their habitats.

It has long been recognized that sediment characteristics greatly influence the structure and diversity of benthic infaunal communities (*Gray, 1974*). Relationships between sediment characteristics and infaunal communities have been studied in a variety of habitats and scales. Sediment grain size, organic content, food abundance, water depth, habitat structure, and salinity have all been shown to influence the composition of infaunal communities (e.g., *Ellingsen, 2002*; *Lindegarth & Hoskin, 2001*; *Maninno & Montagna, 1997*; *Thrush et al., 2001*; *Van Hoey, Degraer & Vincx, 2004*; *Ysebaert & Herman, 2002*). The relative importance of each of these factors varies widely between studies, and these differences may be related to the habitat types, scale, and variability of the environmental factors in the systems being examined. Therefore, developing our understanding of infaunal community composition's relationship to habitat characteristics in habitats for which these relationships have yet to be determined is necessary before larger scale processes can be examined.

Here we examine invertebrate communities at seven intertidal sites along the north central coast of the Gulf of Mexico representing common coastal habitat types at sites spanning a regional scale. Our aim is to explore the relationships between infaunal community structure and sediment characteristics to determine the environmental factors most important in structuring the communities at a regional scale. We focus on the top 5 cm of sediment in these habitats, the depth to which cores were taken by the United States Geological Survey sampling intertidal invertebrates prior to landfall of the Deepwater Horizon Oil Spill in 2010 (*Demopoulos & Strom, 2012*), and the depth most likely to have the highest productivity and influence on higher trophic levels (*Rodil et al., 2008*).

## METHODS

### Field-site description

We collected samples from seven intertidal sites along the northern Gulf of Mexico from Louisiana to Alabama (Fig. 1). Our westernmost site, CAM, was located at Broussard's

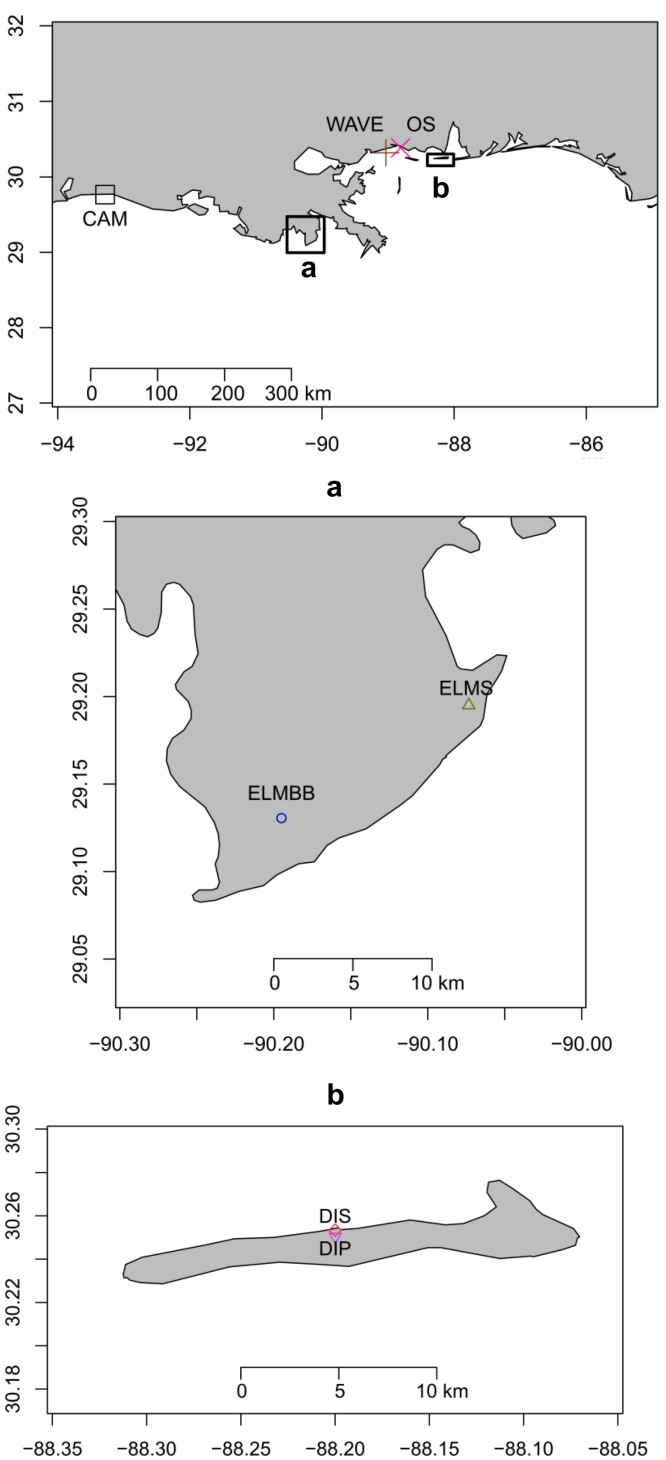

**Figure 1  Map of Northern Gulf of Mexico Coast.** Map of the sampling sites.

Beach (29°45′57.6″N, 93°16′58.8″W), in Cameron Parish, Louisiana. CAM is located near the mouth of the Calcasieu River, which provides freshwater input. The habitat at this site is a sandy beach composed of fine-grained sand with small fragments of shell hash. Two sites, ELMBB (29°7′51.6″N, 90°11′42.0″W) and ELMS (29°11′42.0″N, 90°04′22.8″W) were located on the nearshore barrier island, Elmer's Island, Louisiana. ELMBB is on the back bayside of the barrier island. The habitat is characterized by standing water, which is replenished by tidal inundation. The sediment of ELMBB is composed of mud and fine-grained sand topped by a thin algal mat. ELMS is located on the Gulf of Mexico side of the island, and is a fine-grained sand beach. Freshwater input to Elmer's Island is provided by Barataria Bay and the Mississippi River. Samples from site WAVE were collected at Waveland Beach, Mississippi (30°19′01.2″N, 89°01′58.8″W), a sandy beach near the mouth of Bay St. Louis that is composed of medium-grained sand, some of which is dredged from nearby and periodically added to the beach. OS is a remnant wetland site at Ocean Springs, Mississippi (30°22′58.8″N, 88°48′36.0″W). The sediment in the remnant wetland consists of both mud and medium-grained sand and contains the roots of dead vegetation. Freshwater input to OS is provided by Biloxi Bay. Two more sites, DIS (30°15′10.8″N, 88°11′56.4″W) and DIP (30°15′.00.0″N, 88°11′56.4″W), were on the barrier island, Dauphin Island, Alabama. DIS samples were collected from the bay side of the island, which consists of a sandy shoreline of large-grained sand. The bay side pool habitat, DIP, is characterized by standing water that is replenished by tidal inundation. The sediment at DIP is layered with an algal mat that traps smaller sediment particles lying above a layer of large grained sand. Freshwater input to Dauphin Island is provided by Mobile Bay. All sites have variable salinity dependent on recent weather conditions and the amount of freshwater input provided by nearby rivers and bays.

## Field and laboratory methods

We collected sediment cores from the intertidal area that is a consistently inundated under normal tidal cycles and weather conditions to determine sediment characteristics and infaunal invertebrate community composition at each site. Ten cores were collected from each site, except for ELMS where one sample was lost. Samples were collected between 24 March 2012 and 6 April 2012 using a PVC corer 5 cm deep and 10 cm in diameter. The sediment collected was placed into glass jars which were kept on ice until the samples were returned to the lab. After being returned to the lab, the samples were stored at $-30\,°C$ until processing.

In the laboratory, each sediment core was homogenized and approximately 30 g of sediment from each core for a given site was placed into a composite sample and was stored at $-30\,°C$ for later sediment analysis. The remainder of the sediment from each core was sieved through a 500 μm mesh and the material remaining on the mesh was placed in a mixture of 95% ethanol for preservation and 10% Rose Bengal dye to improve sorting of invertebrates. Using a dissecting microscope, invertebrates were sorted and identified. Infaunal invertebrates were mostly identified to family except for members of Platyhelminthes and Nemertea which were identified to phylum, Bivalvia,

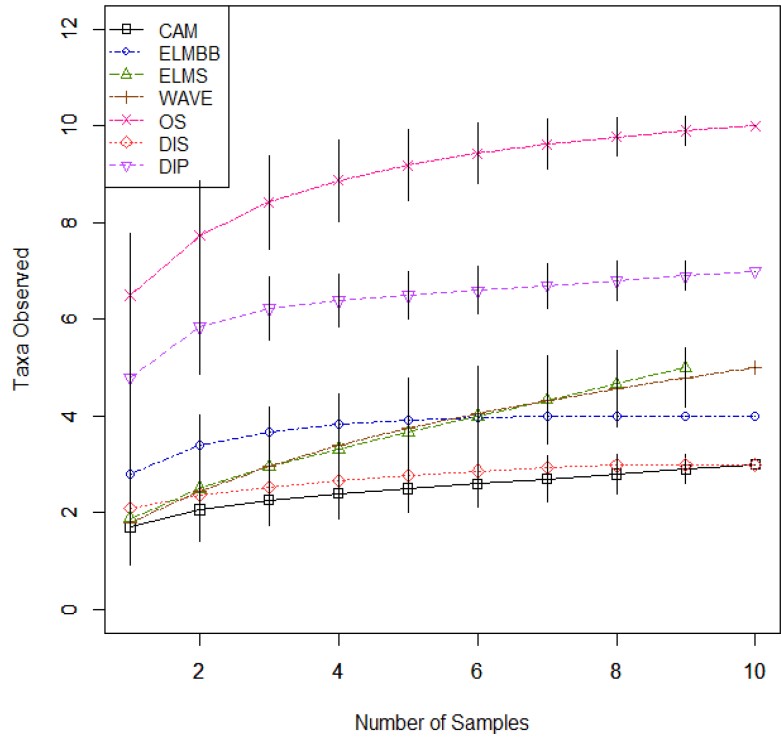

**Figure 2 Taxa accumulation curves for the seven sampling sites.** Taxa accumulation curves calculated from the richness of ten samples at each site, except for ELMS which had only 9 samples.

Gastopoda and Oligocheata which were identified to class and Coleoptera which were identified to order. Invertebrates were not identified to lower levels of classification due to taxonomic uncertainties and to avoid complications arising from using disparate levels of classification.

The composite sediment samples were analyzed for total organic content (TOC), pore water salinity, median grain size and the grain size coefficient of variation. TOC was determined for each site by first drying approximately three grams of sediment in a muffle oven at 60 °C for 14 h and then burning the sample at 450 °C for six hours; TOC was calculated as the percent weight lost between the dry and burned sediment. To obtain pore water salinity, pore water was extracted from the sediment in the lab and salinity was measured using a salinity refractometer. Median grain size and grain size coefficient of variation were measured using an LS 13 320 Laser Diffraction Particle Size Analyzer (Beckman Coulter, Brea, California, USA). Details on the methods used for obtaining the grain size distribution using laser diffraction can be found in *Coblentz et al. (2014)*.

## Statistical methods

Abundance of invertebrates (individuals of each taxon/0.08 m$^2$) was calculated after pooling the 10 sub-samples per site, excepting ELMS where 9 samples were pooled (0.071 m$^2$). For each site, taxa accumulation curves were plotted and visually assessed to ensure that each community had been adequately sampled (Fig. 2). To examine the

relationships between environmental variables and invertebrate community composition, the matrix of square root transformed taxa abundances for all sub-samples was used with non-metric multidimensional scaling (NMDS; *Clarke & Warwick, 2001*) using the Bray-Curtis dissimilarity index to group sites according to community composition. The ordination results were plotted and examined for any clear groupings among sites. Analysis of similarity (ANOSIM) was applied post hoc to test for significant differences between hypothesized groupings. To analyze the relationships between community composition and sediment characteristics, the *bioenv* function was used in the R package *vegan* (*Oksanen et al., 2011*, vegan, v. 2.0-2). This function provides the subset of variables that has the highest Spearman Rank Correlation Coefficient with the Bray-Curtis dissimilarity index (*Clarke & Ainsworth, 1993*). For each of the variables selected by the *bioenv* function, a smooth surface representing the environmental variable was overlaid onto the NMDS using the *ordisurf* function in the R package *vegan* (*Oksanen et al., 2011*, vegan, v. 2.0-2). This procedure develops a visual representation of the relationships between community composition and sediment characteristics. All statistics were performed in R (v2.14.2; *R Development Core Team, 2012*).

## RESULTS

Invertebrates representing five phyla, seven classes, and fifteen families were identified across the 69 sediment cores (Table 1). The invertebrate communities from our seven sites were clustered into four distinct groupings (Fig. 3A; stress = 0.04; ANOSIM, $R = 0.9989$, $p = 0.001$). Our only remnant wetland site, OS, was a unique community and was dominated by tube-building tanaid crustaceans, but had high taxonomic richness including four families of polychaetes, three families of amphipods, and bivalves. 83.6% of the abundance consisted of tanaids, followed by 12.5% corophiid amphipods, and 3% polychaetes. The remainder was composed of gammarid and ischyocerid amphipods, the isopod family Idoteidae, and bivalves. The sandy beach habitat at DIS was also a unique community and was dominated by interstitial oligochaetes and nemerteans, with a low abundance of interstitial platyhelminths. The two back-bay pool sites, DIP and ELMBB, formed a cluster due to similar communities dominated by ephydrid fly larvae, which composed 92% of the abundance at ELMBB and 66.5% of the abundance at DIP. The remainder of the abundance at ELMBB consisted of insect larvae adapted to salt marsh conditions. At DIP, 20.2% of the remaining abundance was composed of insect larvae adapted to salt marsh conditions and 13.2% of the abundance was composed of capitellid polychaetes and oligochaetes. The three remaining ocean-side beach sites, CAM, ELMS, and WAVE, also formed a cluster and had similar communities dominated by haustoriid amphipods (96.5, 97.4, and 97.2% of the abundance respectively) with a small number of bivalves.

The model of environmental characteristics with the highest Spearman Rank Correlation to the dissimilarity matrix included a combination of salinity, median grain size and organic content (*bioenv*, $\rho = 0.67$, Figs. 3A and 4). Salinity was highest in the pool habitats, DIP and ELMBB, and was well over the salinity of seawater, while salinity at the other sites

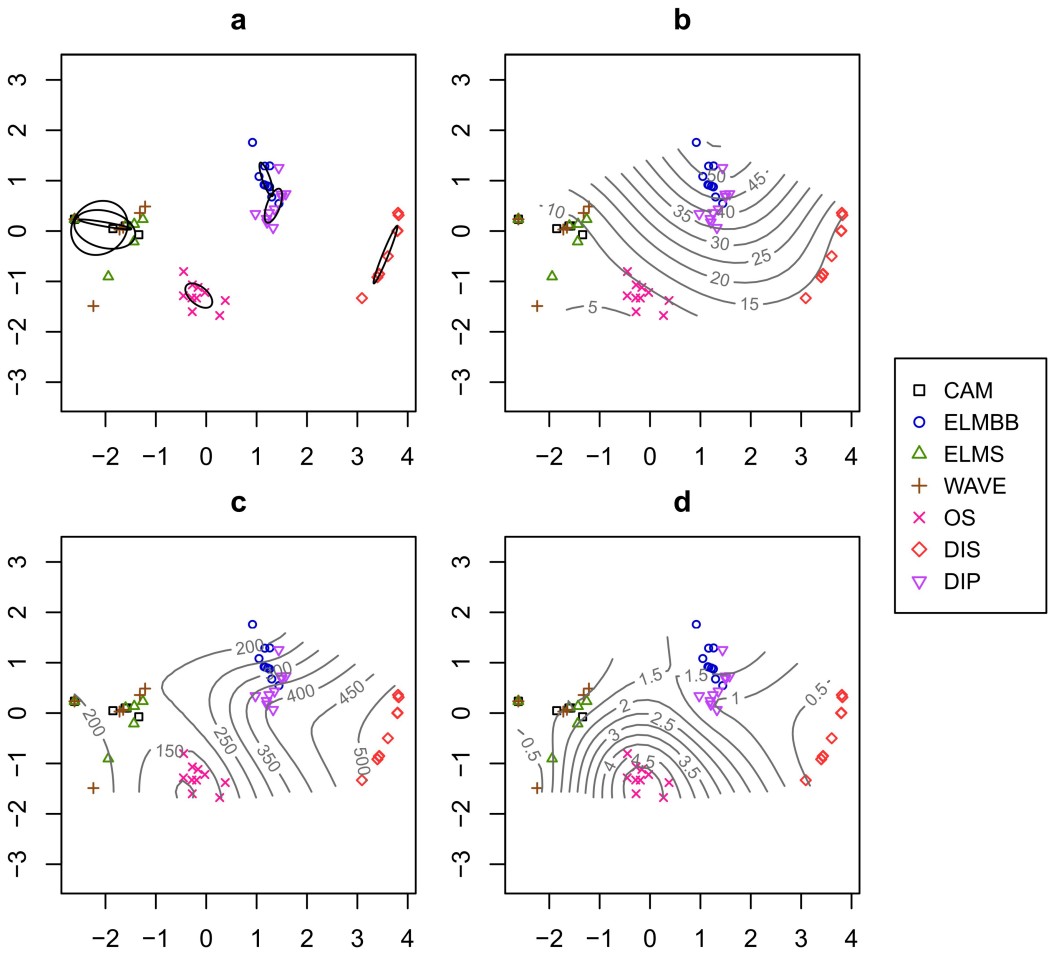

**Figure 3  Community composition across sites.** (A) An NMDS (non-metric multidimensional scaling) plot of sub-samples of intertidal infaunal invertebrate macrofauna collected from seven sites across the northern Gulf of Mexico (stress = 0.04). The samples comprise four distinct communities (DIS; OS; ELMS + WAVE + CAM; ELMBB + DIP; ANOSIM, $R = 0.9989, p = 0.001$). Ellipses represent 99% confidence intervals on the centroids of sites. (B) NMDS plot of sub-samples with a smooth response surface of salinity values overlaid on the NMDS obtained by the function *ordisurf* in the *vegan* package (*Oksanen et al., 2011*, vegan, v. 2.0-2). (C) NMDS plot of sub-samples with a smooth response surface of median grain size values overlaid. (D) NMDS plot of sub-samples with a smooth response surface of total organic content values is overlaid.

ranged from 14 to 5 (Figs. 3B and 4). Sediment median grain size was highest at DIS and lowest at OS, although OS had a high coefficient of variation in grain size (Figs. 3C and 4). TOC was highest at OS and was lower at all other sites, and was particularly low at DIS (Figs. 3D and 4).

## DISCUSSION

Previous studies have related benthic invertebrate community structure to several habitat characteristics with a relatively wide range in the explanatory power of the environmental variables measured (e.g., *Ellingsen, 2002*; *Lindegarth & Hoskin, 2001*;

**Table 1 Community composition of infaunal invertebrates.** Invertebrate abundance per 0.08 m$^2$ except ELMS where abundance per 0.071 m$^2$ is reported.

| Taxa | | | | | | | |
|---|---|---|---|---|---|---|---|
| **Arthropoda** | | | | | | | |
| Amphipoda | **CAM** | **ELMS** | **ELMBB** | **WAVE** | **OS** | **DIP** | **DIS** |
| *Corophiidae* | 0 | 0 | 0 | 0 | 806 | 0 | 0 |
| *Gammaridae* | 0 | 0 | 0 | 2 | 13 | 0 | 0 |
| *Haustoriidae* | 397 | 571 | 0 | 352 | 0 | 0 | 0 |
| *Ischyoceridae* | 0 | 0 | 0 | 0 | 1 | 0 | 0 |
| Tanaidacea | | | | | | | |
| *Leptocheliidae* | 0 | 0 | 0 | 0 | 5410 | 0 | 0 |
| Isopoda | – | – | – | – | – | – | – |
| *Idoteidae* | 0 | 0 | 0 | 0 | 6 | 0 | 0 |
| Insecta | – | – | – | – | – | – | – |
| *Ephydridae* | 0 | 1 | 562 | 0 | 0 | 499 | 0 |
| *Tabanidae* | 0 | 0 | 34 | 0 | 0 | 126 | 0 |
| *Dolichopodidae* | 0 | 0 | 7 | 0 | 0 | 8 | 0 |
| *Ceratopogonidae* | 0 | 0 | 0 | 0 | 0 | 1 | 0 |
| Coleoptera | 0 | 0 | 3 | 0 | 0 | 17 | 0 |
| **Annelida** | | | | | | | |
| Polychaeta | | | | | | | |
| *Nereidae* | 0 | 0 | 0 | 0 | 82 | 0 | 0 |
| *Capitellidae* | 1 | 0 | 0 | 0 | 123 | 67 | 0 |
| *Phyllodocidae* | 0 | 0 | 0 | 0 | 5 | 0 | 0 |
| *Spionidae* | 0 | 1 | 0 | 0 | 4 | 0 | 0 |
| *Orbinniidae* | 0 | 1 | 0 | 1 | 0 | 0 | 0 |
| Oligochaeta | 0 | 0 | 0 | 0 | 0 | 32 | 644 |
| **Nemertea** | 0 | 0 | 0 | 0 | 0 | 0 | 74 |
| **Platyhelminthes** | 0 | 0 | 0 | 0 | 0 | 0 | 6 |
| **Mollusca** | | | | | | | |
| Bivalvia | 13 | 12 | 0 | 4 | 17 | 0 | 0 |
| Gastropoda | 0 | 0 | 0 | 3 | 0 | 0 | 0 |
| **TOTAL** | 411 | 586 | 606 | 362 | 6467 | 750 | 724 |

*Maninno & Montagna, 1997*; *Thrush et al., 2001*; *Van Hoey, Degraer & Vincx, 2004*; *Ysebaert & Herman, 2002*). Here we examined intertidal macrobenthic communities at sites covering common habitat types across a broad geographic region. The communities clustered into four distinct groups, although local variation in abundances of community members was evident. Nevertheless, nearly 70% of variation in the communities between sites was explained by salinity, median grain size, and organic content.

The clustering of communities observed in our data is related to coarse habitat characteristics, which are reflected quantitatively by the sediment characteristics measured. For example, the two pool sites, ELMBB and DIP, clustered together in community composition due to dominance of salt-tolerant insect larvae. This result is not surprising

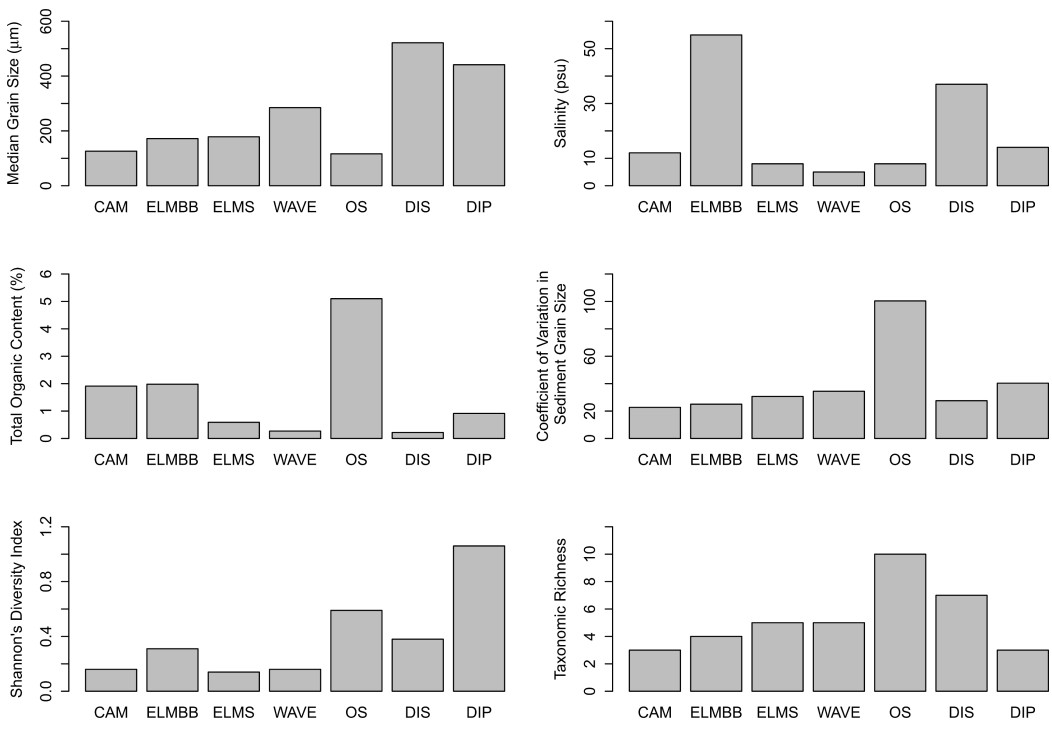

**Figure 4  Summary figure.** Summary of environmental variables, richness and Shannon's diversity measured at each site. A table with exact values can be found in the Table S1.

due to high pore-water salinity measurements in both of the pool habitats, likely from the evaporation of water in the pools. Similarly, presence of the roots of dead vegetation at the remnant wetland site OS provides a unique habitat characteristic among the sites we examined. In turn this habitat characteristic is reflected in the high TOC measurements from the sediment cores at OS. Hence, the three sediment variables included in the model for explaining variation in community composition across sites appear to provide a good quantitative description of the coarse habitat characteristics leading to differences between sites. Similar results have been found in other studies where general sediment characteristics, such as percent mud or silt-clay, explain a large component of the variation in infaunal community composition (*Ellingsen, 2002*; *Ysebaert & Herman, 2002*).

Sediment grain size coefficient of variation was measured, but was not included in the model explaining the largest amount of variation in community composition. However, in another study we showed that sediment grain size coefficient of variation was a significant explanatory factor in the taxonomic richness of these communities (*Coblentz et al., 2014*). This suggests that the variables related to community composition at the regional scale we examined may not be the same as those that are related to diversity in the same communities. In fact, Shannon's diversity was low across all sites and was not associated with any of the environmental variables measured. DIP had the highest evenness of the sites examined, while the remaining sites were dominated by one or a few taxa contributing to relatively low Shannon's diversity values (Fig. 4). The dominance of one or a few

taxa in benthic intertidal habitats has been observed in similar studies and provides an explanation for the low Shannon's diversity values and the variability observed across communities (e.g., *Shelton & Robertson, 1981*; *Ysebaert & Herman, 2002*).

Despite the geographic distances among sites, our data show that sediment characteristics are still able to explain a differences in invertebrate community structure among habitat types. This result is promising for the management of coastal habitats and ecosystems along the northern the Gulf of Mexico. Our results suggest that, rather than needing detailed information on a variety of different environmental variables, knowledge of a few general habitat characteristics is sufficient to predict the "type" of invertebrate community present in a given habitat. The association of community composition to habitat characteristics facilitates a predictive framework of how communities might respond to both natural and anthropogenic changes in habitat characteristics. Although it is unlikely that infaunal invertebrates will be managed directly, their trophic link to ecologically and economically important shorebirds, fishes and invertebrates makes the knowledge of infaunal invertebrate habitat associations an important endeavor (*Gloeckner & Luczkovich, 2008*; *Henkel, Sigel & Taylor, 2012*; *Hsueh, McClintock & Hopkins, 1992*; *McTigue & Zimmerman, 1998*; *Quammen, 1984*).

As coastal ecosystems continue to face pressures due to global climate change and anthropogenic impacts, understanding how communities of organisms are associated with components of their habitats is important to predict and understand how ongoing and future changes are likely to shape the functioning and composition of coastal ecosystems. Here we establish a baseline for this goal in the northern Gulf of Mexico by examining the relationships between infaunal invertebrate communities and the sediment in which they live. To build on this baseline, we suggest that further research examine the ecosystem functions provided by the infauna in different habitat types such as secondary production, bioturbation and nutrient cycling and how these functions are likely to change in response to predicted changes in coastal gemorphology and land use. Research along these lines will help us to understand how future changes will not only affect communities of infaunal invertebrates, but, through their ecosystem functions, their impact on higher trophic levels and the coastal ecosystem as a whole.

## ACKNOWLEDGEMENTS

We thank Dr. Alex Kolker for use of his laboratory to analyze sediment granulometrics, Alex Ameen for assistance in sediment analysis, Kelly Platt and Sarah Romeo for assistance in sorting invertebrates, and other members of CazLab (caz.tulane.edu) for helpful discussion.

### Funding

Funding for this study was provided by the National Science Foundation (Award DEB-1060350, PIs Taylor and Sherry), an NSF REU Supplement (DEB 1112670), a Louisiana Seagrant Undergraduate Research Opportunities Program award (PI Coblentz),

and a Tulane Center for Engaged Learning and Teaching award (PIs Coblentz and Taylor). The funders had no role in study design, data collection and analysis, decision to publish, or preparation of the manuscript.

## Grant Disclosures

The following grant information was disclosed by the authors:
National Science Foundation: DEB-1060350.
NSF REU Supplement: DEB 1112670.
Louisiana Seagrant Undergraduate Research Opportunities Program.
Tulane Center for Engaged Learning and Teaching award.

## Competing Interests

The authors declare there are no competing interests.

## Author Contributions

- Kyle E. Coblentz and Bryan J. Sigel conceived and designed the experiments, performed the experiments, analyzed the data, contributed reagents/materials/analysis tools, wrote the paper, prepared figures and/or tables, reviewed drafts of the paper.
- Jessica R. Henkel conceived and designed the experiments, performed the experiments, contributed reagents/materials/analysis tools, wrote the paper, prepared figures and/or tables, reviewed drafts of the paper.
- Caz M. Taylor conceived and designed the experiments, analyzed the data, contributed reagents/materials/analysis tools, wrote the paper, reviewed drafts of the paper.

## Data Deposition

The following information was supplied regarding the deposition of related data:
Dryad: DOI: 10.5061/dryad.j2c13.

## Supplemental Information

Supplemental information for this article can be found online at http://dx.doi.org/10.7717/peerj.1014#supplemental-information.

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
