# Peer review of "Influence of sediment characteristics on the composition of soft-sediment intertidal communities in the northern Gulf of Mexico"

_PeerJ, doi:10.7717/peerj.1014_

## Round 0.1 · original submission · Minor Revisions

We invite you to submit a revised version of the manuscript that carefully addresses all the critical points of both reviewers.

Reviewer 1 ·

Basic reporting

No Comments

Experimental design

No Comments

Validity of the findings

No Comments

Additional comments

To examine how sediment characteristics and infaunal community composition were related along northern Gulf of Mexico coast, Kyle Coblentz and his colleagues sampled and analyzed intertidal infaunal communities from 7 sites covering common habitat types. They found that nearly 70% of the variations in the composition of the communities could be explained by salinity, median grain size, and total organic content. A couple questions are listed as follows for the authors to consider.

1. In Figure 2, the “Taxa Observed” keeps increasing linearly for samples from ELMS, however, it reaches plateau very quick for samples from ELMBB. How do the authors explain this observation, especially when the two sites are close to each other? I am wondering whether it would be better to include more samples into current study for the sites that “Taxa observed” are far from reaching the plateau?
2. It would benefit the readers if the author could discuss a little bit more about the Shannon’s diversity of samples from different sites.

Reviewer 2 ·

Basic reporting

In this manuscript, the authors investigated the relationship between the sediment characteristics and the composition of soft-sediment intertidal communities in the northern Gulf of Mexico. The authors collected 69 intertidal infaunal communities at seven sites covering common habitat types. These communities were grouped into 4 distinct groups on the basis of faunal composition. The variation in the composition of the communities was explained by salinity, median grain size and total organic content.

Experimental design

All the experiments are well designed. All the experimental procedures were described clearly in this paper.

Validity of the findings

All the findings in this paper were based on solid data and careful experimental design. The results are clearly presented.

Additional comments

Minor concerns:
1. The summary of environmental variables measured at seven sites could be presented as figures like plots instead of table so that the differences between sites are better presented.
2. Phylogenetic tree could be used to show the relationship of isolated invertebrates at each site.
3. Abbreviations for each site just need to be defined at the first appearance. No need to define it every time.

---

## Round 0.2 · accepted · Accept

The authors have made corresponding change according to the reviewers’ comments, I am pleased to Accept this submission.